Differential effects of physical activity on cognitive and motor performance in obese young adults

Sa-nguanmoo Piangkwan piangkwan.s@cmu.ac.th 1 2
Chuatrakoon Busaba 1 2
Keawtep Puntarik 1 2
Thummasorn Savitree 3
Thongsukdee Tanawat 1
Homjan Patimakorn 1
Phetcharat Phumiphat 1
1 Department of Physical Therapy, Faculty of Associated Medical Sciences, Chiang Mai University , Chiang Mai , Thailand
2 Integrated Neuro-Musculoskeletal, Chronic Disease, and Aging Research Engagement Center (I-CARE Center), Department of Physical Therapy, Faculty of Associated Medical Sciences, Chiang Mai University , Chiang Mai , Thailand
3 Department of Occupational Therapy, Faculty of Associated Medical Sciences, Chiang Mai University , Chaing Mai , Thailand
Federolf Peter
Electronic publication date: 2025 Dec 9
Publication date: 2025
Volume: 13
Electronic Location ID: e20481
Received 2025 Jul 25; Accepted 2025 Nov 5
Copyright: ©2025 Sa-nguanmoo et al.
Copyright year: 2025
Copyright holder: Sa-nguanmoo et al.
License: This is an open access article distributed under the terms of the Creative Commons Attribution License, which permits unrestricted use, distribution, reproduction and adaptation in any medium and for any purpose provided that it is properly attributed. For attribution, the original author(s), title, publication source (PeerJ) and either DOI or URL of the article must be cited.
License URL: https://creativecommons.org/licenses/by/4.0/

Keywords: Executive function, Gait speed, Reaction time, Sedentary behavior, Youth obesity

Funding: The authors received no funding for this work.

==============================
The rising prevalence of obesity among young adults presents significant health challenges, particularly due to its adverse effects on cognitive function and physical mobility. This cross-sectional study examined the effects of physical activity on cognitive performance and gait speed in obese individuals aged 18 to 25 years. Seventy-six participants were categorized as either physically active or sedentary based on the Global Physical Activity Questionnaire. All enrolled participants completed the study, and no data were missing. Anthropometric data, including body mass, height, waist circumference, and hip circumference, were collected using standardized procedures. Cognitive assessments included the Trail Making Test, Stroop Color and Word Test, Hand Reaction Time Test, and Logical Memory Test. Gait speed was evaluated using the 10-meter walk test. The physically active group showed significantly better results in logical memory, executive function, and all Stroop test conditions (p < 0.05). No group differences were found in reaction time, Stroop interference score, or gait speed (p > 0.05). These findings suggest that higher physical activity levels are linked to better cognitive performance, highlighting the value of promoting physical activity in young adults with obesity. The lack of observed differences in gait speed and reaction time may indicate that these functions are less sensitive to early changes or require longer periods of inactivity to decline in this population.

Introduction

Obesity is a significant global health issue linked not only to metabolic and cardiovascular diseases but also to cognitive decline (Costache et al., 2023). Emerging evidence indicates that excess adiposity negatively affects cognitive domains such as executive function, working memory, and processing speed, while also contributing to physical limitations, including reduced mobility and slower gait speed (Berbegal et al., 2022; Lentoor, 2022; Vakula et al., 2022). Gait speed serves as a comprehensive, non-invasive biomarker that reflects neuromuscular coordination, cardiovascular health, and cognitive functioning. It predicts functional independence in older adults and serves as an early indicator of declining health in midlife (Rasmussen et al., 2019a; Rasmussen et al., 2019b). The mechanisms connecting obesity to cognitive and motor impairments include chronic low-grade inflammation, insulin resistance, oxidative stress, and cerebrovascular dysfunction (Farruggia & Small, 2019; Huang et al., 2024; Naomi et al., 2023). These concerns are particularly relevant given the rising prevalence of obesity among young adults, a population traditionally considered at low risk for such functional decline. Early identification of modifiable factors, such as physical activity (PA), that can protect against these effects is therefore critical.

PA has well-documented protective effects on both cognitive and physical health. It enhances cerebral blood flow, promotes neurogenesis, improves synaptic plasticity, and reduces systemic inflammation (Ben-Zeev, Shoenfeld & Hoffman, 2022; Latino & Tafuri, 2024). Galle et al. (2023) reported that moderate PA was found to improve both cognitive and physical performance in older adults with initially low activity levels. Additionally, PA interventions have been shown to enhance cognitive function and academic performance in adolescents with obesity (Martin et al., 2018). PA also helps maintain gait speed, which is crucial for physical independence and overall quality of life (Nascimento et al., 2022). While these benefits are well established in older adults, research on the cognitive and motor benefits of PA in obese young adults is still limited. Previous study reported that both total PA levels and cognitive function were significantly lower in adolescents with obesity compared to their non-obese peers (Thummasorn et al., 2022). Importantly, few studies have examined whether PA can simultaneously mitigate both cognitive and motor impairments in young adults at risk due to obesity. Most existing research has evaluated these outcomes independently or within mixed-age populations, leaving a gap in understanding the specific impact of PA in obese young adults.

Moreover, practical motor function indicators such as gait speed and hand reaction time have not been thoroughly examined in relation to habitual PA levels in this demographic. Therefore, the present study aims to evaluate the effects of PA on cognitive performance and gait speed in obese young adults by comparing sedentary and physically active individuals. We hypothesized that the physically active group would demonstrate superior executive function, memory, and cognitive flexibility, as well as faster gait speed and shorter hand reaction times compared to their sedentary counterparts.

Materials and Methods

Study design

This observational cross-sectional study was approved by the Committee for Research in Humans, Faculty of Associated Medical Sciences, Chiang Mai University, in accordance with the Declaration of Helsinki (Approval No. AMSEC-67EX-104). All participants provided written informed consent prior to participation. Participant recruitment and all assessments were conducted between December 2024 and May 2025. As this was a cross-sectional study, data collection for each participant was performed at a single time point.

Study participants

The required sample size for this study was determined using G*Power software (version 3.1). The calculation was based on gait velocity outcomes from a preliminary investigation involving seven participants per group. The mean gait velocities for the physically active and sedentary obese groups were 1.71 ± 0.16 m/s and 1.82 ± 0.17 m/s, respectively. Based on an effect size of 0.66, a statistical power of 0.80, and a significance level of 0.05, a total of 76 participants was required. Eligibility criteria included young adults aged 18–25 years who were classified as obese. Obesity was defined using Asian-specific criteria (BMI ≥ 25 kg/m2) recommended by the WHO Expert Consultation. These thresholds reflect the WHO Expert Consultation’s recognition that Asian populations generally have higher body fat and greater risk of type 2 diabetes and cardiovascular disease at lower BMI levels compared to Western populations (WHO Expert Consultation, 2004). Participants were excluded if they had major comorbidities or conditions that could interfere with testing or confound the results, including acute or chronic illnesses, neurological or musculoskeletal disorders, psychiatric or mood disorders (e.g., depression), and visual or hearing impairments.

Procedure

A total of 76 participants was recruited for the study, with matching based on sex and BMI. PA was assessed using the Global Physical Activity Questionnaire (GPAQ), which records the frequency and duration of moderate- and vigorous-intensity activity across work, transport, and recreational domains (Armstrong & Bull, 2006). In accordance with the GPAQ scoring protocol and WHO recommendations, participants were classified as physically active if they achieved ≥ 600 MET-minutes per week, corresponding to at least 150 min of moderate-intensity or 75 min of vigorous-intensity activity (Bull et al., 2020; World Health Organization, 2010). Eligible activities included moderate-to-vigorous aerobic exercises (e.g., brisk walking, cycling, running), resistance training, high-intensity interval training, and recreational sports, as well as occupational or transport-related activities of equivalent intensity. Light-intensity activities (e.g., casual walking, light household chores) and sedentary behaviors (e.g., sitting, studying, screen time) were not included. According to the MET values derived from the GPAQ, participants were categorized into two distinct groups: the sedentary obese group (n = 38), which reported fewer than 600 MET-minutes per week, and the physically active obese group (n = 38), which reported 600 MET-minutes per week or more. Anthropometric measurements, including body mass, stature, waist circumference (WC), and hip circumference (HC), were recorded. Body composition was assessed using a bioelectrical impedance analyzer (Tanita BC-418; Tanita Tokyo, Japan), including body fat percentage, fat mass, and fat-free mass. Body mass index (BMI) was calculated by dividing body weight (kg) by the square of height (m2). All participants underwent cognitive assessments and a 10-meter walk test to evaluate gait speed. The study protocol is illustrated in Fig. 1.

Figure 1 Flowchart of the study methodology.

GPAQ, global physical activity questionnaire; MET, metabolic equivalent task; WC, waist circumference; HC, hip circumference; BMI, body mass index; BIA, bioelectrical impedance analyzer; TMT, Trail Making Test; SCWT, Stroop Color and Word Test; HRT, Hand Reaction Time test; LM, Logical Memory Test.

Cognitive assessment

Trail Making Test (TMT)

The TMT was used to evaluate executive function and consists of two components: TMT-A and TMT-B. In TMT-A, participants connected numbers sequentially from 1 to 25. In TMT-B, they alternated between numbers and letters in sequence. Performance was measured by the time taken to complete each part. The difference in completion time between TMT-B and TMT-A (B–A) was used as an index of executive function (Tombaugh, 2004).

Stroop Color and Word Test (SCWT)

The SCWT evaluates the ability to inhibit cognitive interference, which occurs when processing one aspect of a stimulus affects the simultaneous processing of another. In this test, the number of correct responses in the word (W), color (C), and color-word (CW) conditions within 45 s was recorded. The interference score (IG) was calculated using the formula: IG = CW − [(W × C)/(W + C)]. A lower IG score indicates greater difficulty with interference inhibition (Scarpina & Tagini, 2017).

Hand Reaction Time (HRT)

The evaluation of processing speed was performed utilizing a HRT test, using a portable electronic timer. Participants were seated and placed their dominant index finger on the right button of a modified computer mouse. Following the presentation of a red-light stimulus, participants were obligated to press the button with maximum rapidity. The average reaction time, quantified in seconds, was computed over the duration of ten trials (Lord, Menz & Tiedemann, 2003).

Logical Memory Test (LM)

The delayed recall component of the Logical Memory (LM) Test was used to assess episodic memory. Participants listened to two short narrative passages read aloud and were instructed to remember as many details as possible. Following a 30-minute delay, they were asked to recall each story as accurately as possible. Higher scores on the delayed recall task indicate better episodic memory performance (Ahn et al., 2019).

Gait speed assessment

The timed 10-meter walk test (TMW) was used to assess gait speed. Each participant began walking from a point 2 m before the designated start line. Timing began as they crossed the start line and stopped at the 10-meter endpoint. The additional 2 m at the beginning and end of the walkway minimized the effects of acceleration and deceleration. The test was conducted twice on the same day, and the average time was used for analysis (Kim et al., 2021).

Statistical analysis

Data were expressed as mean ± standard deviation (SD). Statistical analyses were performed using SPSS version 22.0 (IBM Corp., Armonk, NY, USA). The Shapiro–Wilk test was applied to assess the normality of the data distribution. Independent t-tests were used to evaluate group differences in participant’s general characteristics, cognitive function, and gait speed. The chi-square test was employed to analyze gender distribution. A p-value of less than 0.05 was considered statistically significant.

Results

The general characteristics of the participants are shown in Table 1. There were no significant differences between the sedentary obese and physically active obese groups in age, sex, BMI, body mass, height, WC, HC, waist-to-hip ratio, body fat percentage, fat mass, or fat-free mass. However, the physically active obese group reported significantly higher MET-minutes per week on the GPAQ compared to the sedentary obese group (t (74) = 8.13; p = 0.000; d = 1.86).

Table 1 General characteristics of participants.

Variables	Sedentary obese
(n = 38)	Physically active obese
(n = 38)	P-value	
Age (year)	21.18 ± 1.29	20.73 ± 1.65	0.19	
Gender (Male/female)	13/25	13/25	1.00	
Body mass (kg)	85.01 ± 18.58	83.80 ± 11.43	0.73	
Height (m)	1.65 ± 0.09	1.66 ± 0.08	0.44	
BMI (kg/m2)	30.94 ± 5.73	30.01 ± 3.19	0.38	
Waist circumference	97.97 ± 14.29	95.59 ± 9.95	0.40	
Hip circumference	110.51 ± 12.40	108.87 ± 8.43	0.50	
Waist hip ratio	0.88 ± 0.06	0.87 ± 0.08	0.72	
Fat free mass (kg)	51.47 ± 11.50	52.34 ± 10.20	0.72	
Fat mass (kg)	33.56 ± 13.70	31.46 ± 8.62	0.42	
Body fat percentage (%)	38.85 ± 9.00	37.51 ± 8.37	0.50	
MET-minutes per week	187.36 ± 186.07	2,570.00 ± 1,796.61*	<0.01	
Notes.

Data are represented as mean ± standard deviation (SD). MET, metabolic equivalent task. BMI, body mass index.

* Statistically significant data (P < 0.05).

A comparison of cognitive performance between the sedentary and physically active obese groups is presented in Table 2. The physically active obese group demonstrated significantly better performance in several cognitive tasks compared to their sedentary counterparts. The TMT B-A time was significantly lower in the physically active group than in the sedentary group (t (74) = −2.71; p = 0.008; d =  − 0.70). Similarly, the LM scores were significantly higher in the physically active group compared to the sedentary group (t(74) = 2.06; p = 0.043; d = 0.47). In the SCWT, the number of correct answers in the W, C, and CW conditions were all significantly higher in the physically active group (t (74) = 2.10; p = 0.039; d = 0.34, t (74) = 2.57; p = 0.012; d = 0.34 and t (74) = 2.46; p = 0.016; d = 2.42, respectively). However, no significant differences were observed between the two groups in hand reaction time, IG score, or gait speed (t (74) = 0.28; p = 0.773; d = 0.06, t (74) = 1.14; p = 0.255; d = 0.203 and t (74) = 0.08; p = 0.934; d = 0.019, respectively, Fig. 2). These results indicate that PA may positively influence executive function, memory, and cognitive flexibility in obese individuals, while reaction time and gait speed remain unaffected.

Table 2 Comparison of cognitive performance between the sedentary and physically active obese group.

Variables	Sedentary obese
(n = 38)	Physically active obese
(n = 38)	P-value	
TMT B-A (sec)	51.68 ± 19.54	40.59 ± 15.84*	<0.01	
Logical Memory Test (score)	18.44 ± 6.90	21.31 ± 5.09*	0.04	
Hand Reaction Time Test (sec)	0.246 ± 0.03	0.249 ± 0.05	0.77	
Correct answer in W condition	98.63 ± 13.67	105.76 ± 15.84*	0.03	
Correct answer in C condition	71.44 ± 10.55	78.07 ± 11.83*	0.01	
Correct answer in CW condition	43.97 ± 9.59	49.52 ± 10.07*	0.01	
Interference score	2.65 ± 7.04	4.79 ± 9.07	0.25	
Notes.

Data are represented as mean ± standard deviation (SD). TMT B-A trail making test B-A, W names of colors printed in black, C names different color patches, C names different color-words, CW names color-word, where color-word are printed in an incongruous color ink (name the color of the ink instead of reading the word).

* p < 0.05 vs. the sedentary obese group.

Figure 2 Comparison of gait speed between sedentary and physically active obese groups.

Discussion

Our findings reveal that individuals who engaged in higher levels of PA demonstrated significantly better cognitive performance in executive function, episodic memory, and cognitive flexibility compared to their sedentary counterparts. However, no significant differences were observed in gait speed, hand reaction time, or IG score.

The superior performance on the TMT B-A among physically active participants suggests enhanced executive functioning, including cognitive flexibility and task-switching ability (Fischetti et al., 2024; Shi et al., 2022). This finding aligns with previous research linking PA to improved prefrontal cortex activity, mediated by elevated levels of brain-derived neurotrophic factor (BDNF) and increased cerebral blood flow (Lukkahatai et al., 2025; Tari et al., 2025). The higher LM scores observed in the physically active group further support the cognitive benefits of regular PA, consistent with studies associating moderate-to-vigorous activity with enhanced memory and increased hippocampal volume, particularly in individuals with overweight or obesity (Machida et al., 2022; Migueles et al., 2020). Although physically active participants showed improved performance in all Stroop conditions, the IG value did not differ significantly between groups. The IG score is specifically designed to assess interference inhibition by mathematically adjusting for abilities in word reading and color naming (Scarpina & Tagini, 2017). In the present study, improvements in W, C, and CW conditions occurred proportionally, which may explain the lack of observed enhancement in interference inhibition as calculated by the IG formula. These findings suggest that while PA may improve general processing speed and accuracy, it may not sufficiently enhance the ability to inhibit cognitive interference.

Previous studies have suggested that inhibitory control may require more intensive, targeted cognitive or resistance training interventions to yield measurable improvements (Dhir et al., 2021; Lin et al., 2024). Contrary to our hypothesis, no group differences were observed in gait speed or hand reaction time. This may be attributed to the relatively young age and preserved functional status of participants. In young adults, both neuromuscular and cardiovascular systems are typically well-maintained, which may lead to a ceiling effect that obscures potential benefits of PA on basic motor functions (Tan et al., 2024; Youssef et al., 2024). Furthermore, prior research indicates that complex or fine motor adaptations often require prolonged or highly specific training to manifest (Krzysztofik et al., 2025; Lehmann, Villringer & Taubert, 2022; van Vliet et al., 2023). Another possible explanation for the absence of differences in gait speed is that obesity-related mobility impairments may not yet be clinically evident in early adulthood. Subclinical reductions in neuromuscular efficiency or cardiorespiratory capacity may have been too subtle to affect gait performance, especially in the absence of overt functional decline (Iyer et al., 2024; Koinis et al., 2024).

A key strength of this study is the control of potential confounding variables, including BMI, waist circumference, and body fat percentage, which were comparable across groups. This enhances the interpretation that PA level, rather than body composition, was associated with improved cognitive outcomes. Additionally, while BMI was used to define obesity in this study according to WHO Asian-specific cut-offs, we acknowledge that BMI alone cannot differentiate fat mass from lean mass. To strengthen validity, body composition was also assessed using bioelectrical impedance analysis, and no significant differences were observed between groups in body fat percentage, fat mass, or fat-free mass. Nevertheless, reliance on BMI as an initial screening criterion should be interpreted with caution, and future studies may benefit from incorporating more direct measures of adiposity. However, several limitations should be acknowledged. First, PA was assessed using GPAQ, a self-report instrument subject to recall bias. Future research should incorporate objective measures such as accelerometry for greater accuracy. Second, the cross-sectional design limits causal inference. Longitudinal or intervention studies are needed to determine whether cognitive benefits are sustained over time. Third, while Asian-specific criteria (BMI ≥ 25 kg/m2) were applied in this study based on the WHO Expert Consultation’s recommendation, approximately half of the participants would be categorized as overweight according to international cut-offs (BMI 25–29.9 kg/m2). This difference in classification may limit direct comparisons with studies that use international BMI definitions and should therefore be interpreted with caution when generalizing to non-Asian populations. In addition, participants in this study were undergraduate students recruited from Chiang Mai University, ensuring a relatively homogeneous socio-demographic background with respect to age, educational level, and student status. Nevertheless, potential socio-demographic differences that were not measured may have influenced the outcomes, and this should be considered when interpreting the findings. Fourth, exercise history or training experience was not formally assessed. Although participants reported varying levels of physical activity, the lack of data on training status may limit interpretation of how fitness level or adaptation to exercise influenced cognitive and motor outcomes. Lastly, although several cognitive variables showed significant differences between the physically active and sedentary obese groups, these results should be interpreted with caution due to the potential for alpha error accumulation arising from multiple t-tests. If corrections for multiple comparisons (e.g., Bonferroni adjustment) were applied, only some of the differences might remain statistically significant. Therefore, the present findings suggest a trend toward better cognitive performance among physically active obese individuals rather than definitive evidence of improvement across all domains.

Conclusions

This study suggests that physical activity may have neuroprotective effects in obese young adults, particularly for cognitive performance. However, these findings should be interpreted with caution, and further studies are needed to confirm and extend these results.

Supplemental Information

Supplemental Information 1 STROBE Checklist

Supplemental Information 2 Raw data

Includes all participants assessed for physical activity level, cognitive performance, and gait speed. These data were used in the statistical analysis comparing the physically active and sedentary groups.

The authors would like to sincerely thank the Faculty of Associated Medical Sciences, Chiang Mai University, for providing the facilities and equipment used in this study.

Additional Information and Declarations

Competing Interests

Author Contributions

Human Ethics

Data Availability

The authors declare there are no competing interests.

Piangkwan Sa-nguanmoo conceived and designed the experiments, performed the experiments, analyzed the data, prepared figures and/or tables, authored or reviewed drafts of the article, and approved the final draft.

Busaba Chuatrakoon conceived and designed the experiments, authored or reviewed drafts of the article, and approved the final draft.

Puntarik Keawtep conceived and designed the experiments, performed the experiments, authored or reviewed drafts of the article, and approved the final draft.

Savitree Thummasorn conceived and designed the experiments, authored or reviewed drafts of the article, and approved the final draft.

Tanawat Thongsukdee conceived and designed the experiments, performed the experiments, authored or reviewed drafts of the article, and approved the final draft.

Patimakorn Homjan conceived and designed the experiments, performed the experiments, authored or reviewed drafts of the article, and approved the final draft.

Phumiphat Phetcharat conceived and designed the experiments, authored or reviewed drafts of the article, and approved the final draft.

The following information was supplied relating to ethical approvals (i.e., approving body and any reference numbers):

This study was approved by the Committee for Research in Humans at the Faculty of Associated Medical Sciences, Chiang Mai University (Approval No. AMSEC-67EX-104).

The following information was supplied regarding data availability:

The raw measurements are available in the Supplemental File.

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
