# Peer review of "Differential effects of physical activity on cognitive and motor performance in obese young adults"

_PeerJ, doi:10.7717/peerj.20481_

## Round 0.1 · original submission · Major Revisions

· Academic Editor

Major Revisions

The reviewers generally looked positively on this study.

One important issue to address is the recruitment strategy. How was eit nsured that participants of the two groups came from the same socio-demographic background? E.g. g. were all participants university students? How do you respond to my concern that an unknown confounding socio-demographic factor may affect your results?

Further, BMI>25 is considered "overweight"; "obese" (BMI>30) was only the case for half of your participants. This should be adequately reflected in the paper and in its title.

·

Basic reporting

Well-written manuscript and devoid of verbosity and unnecessary content or contextual and background information.

Sufficient inclusion of summary of key findings from references.

Manuscript is well-structured and figures and tables are professionally and logically arranged. Raw data was also shared.

Experimental design

Experimental design meets the aims and scope of the journal and poses a meaningful, well-defined, and relevant research questions that addresses a knowledge gap in the literature. Methodology was robust and illustrative of research designs from seminal investigations in physical activity and education research.

Validity of the findings

Results are presented and interpreted. Limitations and implications for future research are discussed. This investigation provides heuristic value to PA research and given the age group of participants, carries implications for college students and adult learners.

Additional comments

Strongly consider combining “Anthropometric data were collected” with an existing sentence in the abstract or briefly elaborate on what data points were collected and add it to the sentence. For example: “Anthropometric data, such as body mass, stature, waist circumference, and hip circumference, were collected.”

Lines 63-64: Consider replacing “prior study” with an “earlier investigation” or more succinctly, “Galle and colleagues (2023) reported that moderate PA…”

Lines 215-216: Excellent observation about limitation of the self-reported GPAQ

Lines 218-219: Excellent commentary about recommendations for future research. A randomized controlled trial would be more insightful.

This reviewer opines that the manuscript submission meets all of the review criteria elements: basic reporting, experimental design, and validity of the findings and is publishable with minor revisions.

Overall, the authors did a great job and the findings from this research could potentially bear substantial implications for future research and eventual practice.

·

Basic reporting

Clear and professional reporting of data.
Clear and well-defined references, structure, and appendices.

Experimental design

Paper aligns with the aim and scope of the journal. The research qeustion is clear, and consistent throughout the reserach with standard measurements used for evaluating each component of the sutdy.

Validity of the findings

The paper states that the WHO defines obesity as a BMI of 25; it is actually listed as 30.
https://www.who.int/news-room/fact-sheets/detail/obesity-and-overweight#:~:text=For%20adults%2C%20WHO%20defines%20overweight,than%20or%20equal%20to%2030.

'Physically active' is designed as a time value per week per participant; though, what constitutes that definition for exercise? Is this simply walking, resistance training, HIIT, etc.. There is tremendous value in specifically defining what was inclusive versus exclusive for your definition.

Please elaborate on the mentioned definitions. Additionally, were your participants novices to exercise, and as their body composition/body fat was not evaluated, is there a reason to exclude it? This would be meaningful is establishing a coorelation to specific exercise, and low bodyfat composition compared to high BMI which may provide the same candidate, yet elude to differences at face value alone.

---

## Round 0.2 · Major Revisions

· Academic Editor

Major Revisions

A remaining issue in this study is the alpha error accumulation: a series of t-tests were conducted, but without any correction for multiple comparisons, such as, for example, the Bonferroni correction. Would this correction be applied, then only one of the results could still be interpreted as significant. Please discuss the issue of alpha error accumulation in the manuscript and please word the implications and conclusions more carefully.

Also, please report at least effect sizes with all statistical outcomes. Better would be to report all statistics results in apa-style (t(df)=..; p=...; d=...).

·

Basic reporting

Basic reporting criteria is fulfilled per reviewer, JG.

Experimental design

Experimental design criteria is fulfilled per reviewer, JG.

Validity of the findings

Validity of the findings fulfills criteria per reviewer, JG.

Additional comments

Considerable improvements have been executed in alignment with recommendations from reviewers. Ample clarification was provided in rebuttal.

The progress of the team submitting this manuscript is commendable and greatly appreciated.

However, there are a few recommendations per reviewer, JG:

- Indent Line 77
- Eliminate extra space between "LM" and "scores" (Line 202)
- Elaboration on rationale for WHO recommendation on lowered BMI thresholds among Asian population is recommended (Line 104). This was also noted as a potential limitation. Further, this could potentially limit generalizability of the findings especially among Western populations which was elucidated between lines 235 and 244.

·

Basic reporting

The authors' edits effectively and concisely report their data. The work reads well, and the included figures/tables support the data analysis.

Experimental design

The authors' design aligns with the aim and scope of this journal and contributes to a foundational understanding of their research question. The research question was well defined, and the appropriate attachments are included per IRB requirements, ensuring ethical standards.

Validity of the findings

The measurements selected for this experiment fit the purpose of the study and are supported in prior literature for their validity.

---

## Round 0.3 · accepted · Accept

· Academic Editor

Accept

I am satisfied with the revisions and believe the manuscript is ready for publication.

·

Basic reporting

All elements of basic reporting have been fulfilled.

Experimental design

Experimental design, data analysis, and reporting is in alignment with aims and scope of journal and diligence has been taken to address and remediate recommendations for revision.

Validity of the findings

Findings offer heuristic value and bear immense future research and eventual practice implications. Data has been presented and corrections have been clarified per the request of one of the reviewers.

Additional comments

N/A